# The Effect of Life Stages on the Experience of Those Who Have Received an Unexpected and Violent Death Notification: A Qualitative Study

**DOI:** 10.3390/ijerph21070915

**Published:** 2024-07-13

**Authors:** Diego De Leo, Andrea Viecelli Giannotti, Nicola Meda, Martina Sorce, Josephine Zammarrelli

**Affiliations:** 1Australian Institute for Suicide Research and Prevention, Griffith University, Mt Gravatt, QLD 4122, Australia; 2Slovene Centre for Suicide Research, Primorska University, 6000 Koper, Slovenia; 3De Leo Fund, Research Division, 35137 Padua, Italy; 4Italian Psychogeriatric Association, 35137 Padua, Italy; 5Department of Neuroscience, University of Padova, 35121 Padua, Italy

**Keywords:** traumatic death, death communication, notification of death, suicide, road accident, survivors, age groups

## Abstract

Background: How individuals are informed of the traumatic loss of a loved one can influence their grieving process and quality of life. Objective: This qualitative study aimed to explore, through thematic analysis, how life stages might influence the experience and feelings of those who have received communication of a traumatic death from police officers or healthcare professionals. Method: Recruited through social networks and word of mouth, 30 people participated in the study. Subjects were divided into three groups according to age (Group 1: ten participants aged between 20 and 35 years; Group 2: ten participants aged between 45 and 55 years; and Group 3: ten participants aged 60 and over). Participants completed an ad hoc questionnaire online. Atlas.ti software 8 was used to perform thematic analysis. Results: The three age groups had the following four key themes in common: (a) emotional reactions; (b) subjective valuation of the notification; (c) support; and (d) needs. Subtle differences emerged between age groups; yet the quality of the reactions and main themes did not vary greatly between the groups considered. Conclusions: The communication of an unexpected and violent death seems to provoke rather similar effects in survivors of different life stages. A few differences were noted in sub-themes (increased need for professional training in younger recipients; absence of suicidal ideation in older adults); perhaps quantitative designs could provide further details in future investigations.

## 1. Introduction

The communication of death is a delicate task, usually handled by professional figures such as doctors, nurses, police officers, or even religious figures [1]. The way a death is notified represents a moment that is seen as an important phase of the grieving process, as it has a significant influence on the emotional and behavioral responses of the recipient [2]. This is especially true when the person receiving the notification of death is not mentally prepared for the event, as in the case of traumatic deaths occurring unexpectedly due to external and violent causes (suicide, homicide, accidents, and natural disasters) [3].

The impact caused by the notification of death on the recipient is influenced by several variables, such as the words and expressions used by the notifier, the characteristics of notifiers themselves, where the communication takes place, and how it is performed (e.g., in presence, via phone or email, etc.) [3]. For example, studies conducted within intensive care units showed that the experience of death notification was also influenced by certain characteristics of the recipient, such as age, gender, presence of history of psychiatric treatment, marital status, and whether the recipient had children [4].

Concerning age, each phase of life is associated with different levels of cognitive functioning [5], different developments in personality traits [6], and different emotion regulation strategies [7]. Understanding how these variations affect the bereavement process through different life stages could be important for providing the appropriate support. For example, in early childhood, the concept of death is not comprehended. Signs of distress are somatic and behavioral, and crying, changes in eating or sleeping patterns, and increased clinginess are common. The beginning of an understanding of the irreversibility of death happens typically around the ages of 6–7 years [8]. It is during adolescence that individuals develop a more mature understanding of death and its implications. Grief may be expressed through mood swings, withdrawal, risky behaviors, or academic difficulties. In all these life stages, immature emotion regulation strategies might be moderators of the expression of grief [7].

In adulthood, individuals have a full understanding of death, and a vast array of reactions to grief can develop such as sadness, anger, anxiety, and reevaluations of life goals and relationships. In late adulthood, individuals typically possess a profound understanding of mortality and may have experienced numerous losses. Grief in late adulthood is often intertwined with life reflections, increased isolation, and potential physical or cognitive decline [9,10].

In his theory of psychosocial development, Erikson divided the entire span of life into eight different stages, each associated with a precise and critical phase of the development of the personality as a whole [11]. The aim of this step-by-step process is for the person to reach old age psychologically “intact/complete” and this is only possible if, as one gradually comes across the various tasks of life, one manages to acquire the ability to develop the mechanisms and strategies that allow to face and manage the changes, losses, and traumas that life presents to individuals. This mechanism of personality evolution is a path that begins at birth and continues without interruption until the moment of death. Therefore, the experiences lived in each stage can influence the individual’s well-being in subsequent stages [11].

By analyzing the reports of survivors, the aim of this article was to qualitatively investigate the relationship between bereavement process from a traumatic death notification and life stages. In doing this, we did not have a strong a priori hypothesis, as no previous research appears to have explored this relationship. However, we expected younger individuals to report more intense externalizing reactions in comparison to older individuals, who should have matured and gained a greater level of resilience and coping strategies potentially useful in mitigating the stressful impact of the communication of an unexpected and violent death. In accordance with Erikson’s theory, in fact, older adults should have achieved greater identity integrity and coping capabilities [11], due to the progressive overcoming of the different developmental life stages.

## 2. Methods

This study was part of a broader research project called "IRIS," designed to examine the communication of unexpected and violent deaths from both the perspective of the notifiers (often police officers and healthcare workers) and that of the recipients (family members and friends). This research project was approved by the Ethics Committee of the University of Padua (no. 3878), and informed consent was obtained from all subjects involved.

Participant recruitment was conducted through snowball sampling, using the authors’ network of contacts, word of mouth, and social media (Facebook and Instagram), including the involvement of the NGO De Leo Fund. For the present study, we considered 30 survivors, all family members of victims of violent and sudden deaths, such as suicide and road accidents. All individuals responded to an online questionnaire created ad hoc with 12 open-ended questions. Before administering the questionnaire, a performance pre-test was conducted on a small group of individuals, including five suicide survivors, to assess the efficiency of the investigative procedure.

The questionnaire was designed to avoid re-traumatization, considering the risk of suicide of the special population of survivors [12]. It was structured such that participants could freely withdraw without the need for any explanation if they found the questionnaire to be too intrusive. We aimed to analyze the process of notification of unexpected and violent death in relation to the following three age groups: a sample of young adults, one of middle-aged people, and one of older adults. We were able to recruit ten survivors from 20 to 35 years old (Group 1), ten people from 45 to 55 years old (Group 2), and ten over 60 years of age (Group 3). We considered the three groups selected as grossly representative of three different moments in the life cycle. Out of the 30 people, 4 were males and 26 females. This disproportion is justified by the fact that, in Italy, the suicide ratio for males/females is 3.5/1 and the road accident ratio is 5/1 [13]. Consequently, women are overrepresented among traumatic death survivors.

We adopted a qualitative interpretative design to conduct a thematic analysis of the participants’ written reports of their reactions to the process of a death notification. By delving into the ways individuals process grief and adapt to loss across the different age groups, we aimed to contribute to a deeper understanding of the diverse experiences of bereavement.

Data collection took place from January to July 2021. We considered that ten individuals for each age group could involve theoretical saturation, meaning that further data would not provide further information.

We employed Atlas.ti 8 software for thematic analysis [14]. Thematic analysis is one of the most used methods in qualitative research, as it allows you to effectively identify and analyze recurring themes in the research data, in order to organize them and possibly formulate hypotheses. Through the Atlas.ti 8 software and by adopting an inductive approach, the data collected were labeled in order to identify patterns or regularities.

Two researchers (AVG and JZ) familiarized themselves with the data obtained from the questionnaires. The responses were read multiple times, and the initial impressions were noted. The same researchers generated initial codes by collecting salient resulting data for each code. Subsequently, the codes were grouped into themes, and these themes were reviewed. In the next step, a third researcher (DDL), external to the initial part of the study, reviewed the themes and, together with the first two researchers, constructed the final themes and labeled them. At the end of the examination, a final report was produced with a description of the obtained results.

## 3. Results

The results were grouped into two categories. The first concerned the main aspects relating to the traumatic loss; the second considered the themes that emerged from the analysis.

### 3.1. Information Relating to the Loss

The following three variables were analyzed: cause of death, type of relationship with the deceased, and time from death. The three variables considered appeared to be sufficiently homogeneous among all three age groups.

#### 3.1.1. Cause of Death

The causes of death reported by participants were suicide and road accidents. Specifically, in G1 and G3, they were equally distributed in each group (five cases of suicide and five cases of road accidents); in G2, four deaths occurred due to road accidents, while six deaths were due to suicide (see Table 1).

#### 3.1.2. Degree of Kinship

In the younger group (20–35 years old), two were survivors of their children, four were brother/sister, and four were wife/husband.

In the group aged 45 to 55, three were the parents of dead children, two were brother/sister, and five were wife/husband.

In the older group, four participants were parent survivors, three participants were brother/sister, and three participants were wife/husband.

#### 3.1.3. Time from Death

The following three time series in which the loss of a loved one occurred were taken into consideration: from 2016 to 2020, from 2011 to 2015, and up to 2010.

Between the years 2016 and 2020, three participants of groups G1 and G3 reported suffering their loss, while this was the case for five participants of group G2.

Four participants from G1 and G3 reported having lost their loved one in the period from 2011 to 2015, while this was the case for two participants of G2.

For deaths that occurred up to the year 2010, three participants of each group of the study (G1, G2, and G3) lost a loved one.

### 3.2. Key Themes

The following four themes were identified, along with the corresponding sub-themes (see Table 2): (1) Emotional reactions; (2) Subjective assessment of the notification; (3) Support; and (4) Needs.

#### 3.2.1. Emotional Reactions

This theme was articulated into the following three specific sub-themes: (a) immediate impact, (b) reactions after the notification, and (c) reactions towards the notifier.

##### Immediate Impact

All study participants reported a severe emotional impact experienced during the notification process. The predominant experience for all three groups was mostly represented by feelings of confusion, shock, and disbelief. 

*“I was in shock, disbelief... I felt emptied of every emotion, as if I had been petrified”* (said a person in G1, but similar words were used by subjects of all groups).

Psychological pain and a sense of emptiness and emotional detachment were also frequent reactions. 

*“It’s inhumane pain. I was sure I would die of a broken heart shortly after. I started screaming and crying desperately. I lashed out at the doctor out of anger. But it wasn’t his fault. I wanted my son back”* (a subject in G2).

Experiences of intense anguish and panic were often reported in G3, as well as in G2, but were not referred to by people in G1. 

*“Panic. I was trembling like a leaf and felt my heart pounding out of my chest”* (a subject in G2).

While thoughts of death/suicidal ideation were not reported in the sample aged 60 or over, they were reported by individuals in G1 and G2.

##### Reactions after the Notification

Reactions following the notification included confusion and disbelief (G1 = one individual; G2 = two individuals) and were particularly frequent in participants of the third group (G3 = six subjects), who did not experience the sense of emptiness and emotional detachment present in the other two groups (G1 = two individuals; G2 = one individual). 

*“People passing by. Nurses around. Everything was too much and too little. 5 minutes of hell for me and my husband, while the rest of the world around continued to live”* (a person in G2).

*“Much confusion. I didn’t understand. I remember looking at the doctor, but the background around her was blurry”* (a person in G1).

Flashbacks and intrusive thoughts regarding the moment of death notification were present across all groups but were particularly noticed in the youngest group.

##### Reactions towards the Notifier

The following reactions towards the notifier were quite homogeneous for survivors of the first and second groups: anger and despair, together with feelings of respect: 

*“If my husband hadn’t intervened, I would have been violent towards the doctor”* (one person in G2).

*“I shook his hand and thanked him”* (one survivor in G1).

Reactions in the form of feelings of detachment, confusion, and apathy were very frequent in all groups.

*“I had no reaction towards the doctor. I just remember that my mother, father, and I hugged each other.”* (one individual in G1).

#### 3.2.2. Subjective Assessment of the Notification

##### Communication Quality

Several study participants described the quality of the communication they received as characterized by negative elements (lack of clarity, confusion, and incomplete information).

*“In the beginning, everything was confused; they told me I had to wait for checks”* (one person in G2).

However, some other survivors also recognized positive elements, perceiving the closeness of the notifier and appreciating the transparency of the information and the humanity with which the death was communicated.

*“[I was notified] with as much humanity as possible but at that moment, I was in shock and what they told me felt unreal”* (one person in G1). 

Answers concerning this category appeared as homogeneous and evenly distributed across the three groups.

##### Criticisms about the Communication

Some people in Group 1 were the only ones to identify critical elements regarding the location of the communication, underlining the lack of privacy and the hurried communication from the notifier. Two people in G2 expressed disappointment in generic terms. All three groups expressed criticisms regarding the coldness, rudeness, embarrassment, feigned empathy, and inappropriate choice of personnel for the notification. 

*“Rudeness and extreme roughness in oral communication (move from here; don’t break down; we can’t tell you anything; get out of the room; don’t interfere with our work; you’re annoying now; etc.)”* (persons in G3).

However, a considerable number of participants did not identify or recall criticisms.

#### 3.2.3. Support

Overall, the most relevant source of support for all three groups was represented by family and friends.

*“My family... my aunt, my sister, and my mother tried to comfort each other”* (a person in G1).

Some people looked for formal psychological support, as well as strategies such as writing a book, having a pet, and praying.

*“I worked on myself, I understood a lot, that they won’t come back. And I wrote a book in 20 days, self-produced with great success. Now, I’m writing another one that will be released on June 26th, two years after my son’s rebirth in the beyond.”* (one person in G1).

#### 3.2.4. Needs

Virtually all the people denounced the need for a better death communication, performed by professionally trained notifiers. Survivors from Group 2 and 3 expressed the need for greater closeness, calmness, clarity, and information about the availability of support services.

*“[I would have needed...] Support from a person prepared for this type of trauma (healthcare worker or other).”* (a person in G3).

The need for psychological support was more pronounced in Group 1. The need for social support was also marked in G1 but was scant or absent in the other two groups.

## 4. Discussion

The communication of traumatic death is a sensitive task that holds significant importance in the bereavement process due to its profound effect on the emotional and behavioral responses of the recipient [15,16]. Considering that different life stages are associated with distinct cognitive functioning and emotion regulation strategies [17,18], this paper aimed to qualitatively investigate the relationship of life stages with the experience of death notification and bereavement following a traumatic death. 

We recruited 30 individuals who had experienced the traumatic loss of a loved one. The participants in this study had lost a spouse (40%) or a child (30%). The distribution of relationships varied slightly across age groups, possibly reflecting the diverse social networks and familial structures characteristic of different life stages. 

The most common manner of death were suicide (53%) and road accidents (47%). There was quite a similar distribution of these causes among younger and older individuals.

Time distance from the loss across the considered periods (i.e., up to 2010, 2011–2015, 2016–2020) was relatively consistent across all age groups. 

We adopted a qualitative interpretative design to conduct a thematic analysis of participants’ written reports of their cognitive–behavioral responses to the process of death notification.

Traumatic death experience per se can be overwhelming, irrespective of age, gender, socio-demographic status, or personal history. On top of that, traumatic death notification can be an additional source of distress and pain. Bearing in mind the overwhelming nature of the traumatic loss of a loved one, here, we sought to evaluate how the ways of processing grief and adaptation could vary according to the age of the survivors. We categorized participants into three age groups—from 20 to 35 years old, from 45 to 55 years old, and over 60. 

Thematic analysis identified the following four themes: emotional reactions, subjective evaluation of the notification, support, and needs across all age groups. People aged 20–35 (Group 1) reported confusion and shock, similarly to the other age groups, and remarkable detachment/feeling of emptiness upon receiving the notification of death. After the notification, participants reported flashbacks and intrusive thoughts as predominant themes. The prevalence of flashbacks and intrusive thoughts among younger individuals suggests a potentially higher susceptibility to trauma-related symptoms in this age group [19,20]. This could be due to the interaction among the traumatic nature of the deaths, the secondary traumatization of notification, and suboptimal coping mechanisms, as is expected in young individuals [11,21,22]. However, further studies would be needed to elucidate the exact mechanism through which this happens, or if these feelings are possible predictors of psychopathology. This group mainly reported feelings of detachment, as well as both positive and negative reactions towards the notifier. When asked to subjectively evaluate the communication with the notifier, some participants of this group were the only ones mentioning the location of the communication (which lacked privacy, an aspect that is often overlooked) [23]. People aged 20 to 35 also reported the need for professional and social support more than the other groups.

People aged 45–55 (Group 2) reported confusion and shock; anguish was also mentioned, as well as thoughts of death at the time of notification. In this study, we did not distinguish between passive and active suicidal ideation. Thoughts of death have been reported to be present in more than 90% of people bereaved by traumatic death even 3.5 years after the loss [23]; active suicidal ideation is less frequent, although often reported [24,25]. The theme of flashbacks and confusion after notification was less frequently reported in this group than in that of younger participants. Feelings towards the notifier overlapped with those of Group 1. The theme of support was connotated by expressing the need for the presence of family members and peers. This group also expressed the need for more clarity during communication as well as information on help services.

Older adults (over 60 years of age—Group 3) reported confusion and shock at the time of notification, similarly to the other groups, but anguish was more frequently reported in this group than in other age groups. Notably, no mention of void/emptiness was made. After notification, feelings of confusion and disbelief were predominant if compared to the other age groups. Interestingly, in this group the only feelings expressed towards the notifier were detachment and apathy. The themes of support and needs overlapped with those of the group of middle-aged participants. No suicidal ideation was reported from this group of people.

The coping mechanisms employed by each age group differed. Younger individuals appeared to rely more on seeking professional and social support to cope with the emotional impact of death notification, whereas middle-aged and older adults seemed to prioritize the presence of family members and peers for support. This could be attributed to variability in the social networks and support systems across the different life stages.

Despite differences in the emotional responses, the subjective evaluation of the communication received did not seem to vary across age groups. Participants highlighted both negative aspects, such as lack of clarity, and positive elements, like perceived closeness and humanity. However, individuals from all groups criticized the coldness and rudeness of the communication. Although traumatic death notification often leads to painstakingly remembering the details of the communication [26], a substantial proportion of participants (18/30) had no criticism to report or did not remember negative/positive aspects of the communication. It could be argued that poor memory recall is linked to the effect of trauma [27]. However, it is not possible to discern the effect of the traumatic loss from a traumatizing communication of the loss itself. On this matter, a hypothesis was formed to test is if void and detachment were secondary to suboptimal emotion regulation skills or to a limited familiarity with death. Moreover, we noticed that younger participants reported stronger feelings of detachment and flashbacks. In line with the theories regarding the consolidation of traumatic memories, we propose that the propensity to experience detachment/dissociation after traumatic events or traumatizing communication could be linked to impaired memory recall of details (more than half of the participants had no criticism regarding the notification) and consolidation of salient (and later experienced as intrusive) memories [28,29].

## 5. Limitations

Our study relied on written reports, possibly introducing bias. The qualitative design of this study involved a small sample size and, although comparisons were made among different age groups, generalizability was out of the question. Also, causal conclusions about age-related differences were not possible. Notably, 9 out of 30 traumatic deaths occurred 10 years before this study was conducted. Although the time of traumatic death and time of recruitment were included in the same age of life for most of the participants, it should be noted that this might have introduced a strong recall bias. Therefore, in some cases, this study might have assessed how the stage of life affected the recall of the perceived reactions. Moreover, the study did not differentiate between passive and active suicidal ideation, which could be a point for future research. Lastly, it would have been important to assess the emotional, social, and practical support that survivors could receive from their families, acquaintances, and friends, as social network structure and connections vary in different life phases.

## 6. Conclusions

This qualitative study examined how different life stages may influence the experience of traumatic death notification. We identified four themes across age groups, including emotional reactions, notification evaluation, support, and needs. Younger individuals showed heightened trauma-related symptoms, while middle-aged and older adults expressed anguish and confusion. We found the following age-related coping strategies variations: younger adults reported a need for professional support, while older adults reported a greater need for family support. These differences may stem from different cultural environments and social support systems across ages.

Future studies could employ mixed-methods research for a more comprehensive view of traumatic death notification reactions and impacts. Longitudinal research could also clarify the long-term effects of death notification, possibly promoting properly grounded recommendations for different age groups in the difficult field of death communication.

## Figures and Tables

**Table 1 ijerph-21-00915-t001:** Information relating to the losses: date of death, degree of kindship, and cause of death.

	Date of Death			Degree of Kindship			Cause of Death	
	2016–2020	2011–2015	Up to 2010	Parent	Brother/Sister	Wife/Husband	Car Accident	Suicide
G1 (20–35 years old)	3	4	3	2	4	4	5	5
G2 (45–55 years old)	5	2	3	3	2	5	4	6
G3 (over 60 years old)	3	4	3	4	3	3	5	5

**Table 2 ijerph-21-00915-t002:** Summary of themes and sub-themes.

Theme	Sub-Theme	G1: 20–35 y	G2: 45–55 y	G3: >60 y
**Emotional reaction**	**Impact**	Confusion, shock, anguish, void/detachment ++	Confusion, shock, Anguish + Void + **Thoughts of death/suicide**	Confusion, shock, Anguish ++ **No void**
	**Feelings after notification**	Confusion Flashback ++	Confusion Flashback +	Confusion/Disbelief +++ Flashback +
	**Feelings towards notifier**	Detachment Positive/Neg. affect		Detachment **Apathy**
**Subjective evaluation**	**Quality**	No apparent difference		
	**Criticism**	Many did not remember. Some reported coldness. Mentioned wrong location		
**Support**		Asked for presence of family members and peers ++		
**Needs**		Psychological and social support +++	Mentioned clarity and information on support services	

Pivotal feelings and thoughts of participants according to age group. “+” symbol indicates the frequency of that feeling/thought/experience in the age group, with maximum frequency expressed by “+++”. Feelings/thoughts/experiences in bold appeared as unique to that specific age group.

## Data Availability

The data presented in this study are available upon reasonable request to the corresponding author. The data are not publicly available, due to their confidential nature.

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
