# Peer review of "The Effect of Life Stages on the Experience of Those Who Have Received an Unexpected and Violent Death Notification: A Qualitative Study"

_ijerph, 2024, doi:10.3390/ijerph21070915_

Round 1

Reviewer 1 Report

Comments and Suggestions for Authors

This is a good article about the impact of communication of death. The introduction is too short. I suggest to improve and describe the aim of the research. 

- Methods: How was Atlas 8 used? Please add informations

- A table resuming the results should be added

- Too many self citations. They should be removed.

Comments on the Quality of English Language

Minor issues detected

Reviewer 2 Report

Comments and Suggestions for Authors

The submission summarizes the results of a qualitative study on how individuals who experienced the loss of a loved one from an external cause of mortality recall the notification of the death and various aspects of their response to it. The authors argue that the literature does not include an examination of how stage of life may be associated with the response process. The manuscript suffers from several major limitations, and the study design and data collection make it unlikely that the authors can satisfactorily overcome these limitations. I will highlight the major ones below:

1.       This study cannot analyze the association between stage of life and reaction to the death notification. The study sample is grouped according to age at data collection, but the many of the traumatic deaths occurred several years before. This study may, at best, be able to assess how current stage of life affects the recall of perceived reactions?

2.       The authors indicate the use of a thematic analysis, but the presentation of the results is consistent with content analysis. They emphasize raw counts of individuals who respond on certain ways and then attempt to interpret differences in the numbers as if the study were quantitative in orientation.

3.       The authors describe an ad hoc data collection form. It is assumed that there was no ability to probe for clarification of participant responses. It is unclear to what extent the themes that emerged are a function of what questions were asked.

4.       There is no connection to theory. Why would stage of life be a relevant factor here? How would you expect stage of life to affect the process?

Comments on the Quality of English Language

Language quality was fine. General editing is all that is required.

Round 2

Reviewer 2 Report

Comments and Suggestions for Authors

The authors have addressed the critiques and provided appropriate limitations to the interpretation of the study. 

Comments on the Quality of English Language

A few errors but they do not impede readability.